# CoVLM: Composing Visual Entities and Relationships in Large Language Models Via Communicative Decoding

**Junyan Li**
UMass Amherst

**Delin Chen**
Wuhan University

**Yining Hong**
University of California, Los Angeles

**Zhenfang Chen**
MIT-IBM Watson AI Lab

**Peihao Chen**
South China University of Technology

**Yikang Shen**
MIT-IBM Watson AI Lab

**Chuang Gan**
UMass Amherst and MIT-IBM Watson AI Lab

## Abstract

A remarkable ability of human beings resides in compositional reasoning, *i.e.*, the capacity to make "infinite use of finite means". However, current large vision-language foundation models (VLMs) fall short of such compositional abilities due to their "bag-of-words" behaviors and inability to construct words that correctly represent visual entities and the relations among the entities. To this end, we propose CoVLM, which can guide the LLM to explicitly compose visual entities and relationships among the text and dynamically communicate with the vision encoder and detection network to achieve vision-language communicative decoding. Specifically, we first devise a set of novel communication tokens for the LLM, for dynamic communication between the visual detection system and the language system. A communication token is generated by the LLM following a visual entity or a relation, to inform the detection network to propose regions that are relevant to the sentence generated so far. The proposed regions-of-interests (ROIs) are then fed back into the LLM for better language generation contingent on the relevant regions. The LLM is thus able to compose the visual entities and relationships through the communication tokens. The vision-to-language and language-to-vision communication are iteratively performed until the entire sentence is generated. Our framework seamlessly bridges the gap between visual perception and LLMs and outperforms previous VLMs by a large margin on compositional reasoning benchmarks (*e.g.*, $\sim 20\%$ in HICO-DET mAP, $\sim 14\%$ in Cola top-1 accuracy, and $\sim 3\%$ on ARO top-1 accuracy). We also achieve competitive performances on traditional vision-language tasks such as referring expression comprehension and visual question answering [1].

## 1 Introduction

A remarkable ability of human beings resides in compositional reasoning: the capacity to construct an endless number of novel combinations from a finite set of known components, *i.e.*, "infinite use of finite means" (Chomsky, 1965; 1957; Montague, 1970). As depicted in Figure 1, for someone who has never witnessed a scene where a person sits on a sulky, it's not hard to render this conclusion by combining the known components - "man", "is sitting on" and "sulky". Compositionality is omnipresent in the language such that a sentence is made up of words like nouns ("man") and verbs ("sit"). It also exists ubiquitously in vision so that we could easily detect visual entities such as the person and the sulky, composed with relationships like "sit on". It's believed by cognitive scientists that the meaning of a sentence lies in the interaction between an utterance and external situations

---

[1]Project page: https://vis-www.cs.umass.edu/CoVLM/

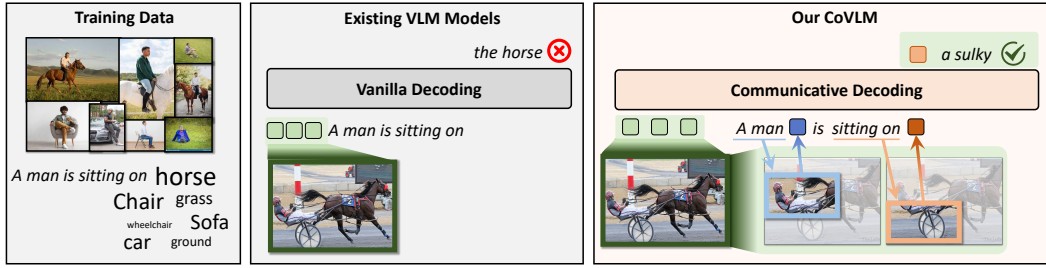

Figure 1: Comparison with existing VLMs. Previous models take in a whole image as input, impairing the compositionality of VLMs. Our CoVLM inserts communication tokens into the LLM after visual entities/relationships to enable language-to-vision and vision-to-language communication, improving compositionality to a large extent.

that can be perceived - the meaning of a noun phrase is linked to a visual entity, and the meaning of a verb phrase is linked to a relational property (Janssen & Partee, 1997). From the meanings of the subject, verb phrase, and object, the sentence is built in a systematic and compositional way.

Current Vision-Language Models (VLMs), however, tend to fall short of such compositional abilities (Ma et al., 2023; Cascante-Bonilla et al., 2023; Doveh et al., 2022; Zhao et al., 2023). As noted by recent works, deficiency of compositionality in these VLMs is likely due to the hypothesis that they behave like "bag-of-words" (Yuksekgonul et al., 2022) - that they merely memorize by rote the frequent co-occurrences of words, but fail to construct words that could correctly represent objects and the relations between objects. Previous works (Zhao et al., 2023; Cascante-Bonilla et al., 2023) have shown that VLMs struggle a lot when relationships are involved. We can also come to this conclusion from Figure 1, in which the models utilize the shortcut learned from pre-training that "a man sits on a horse" appears frequently and there's a man and a horse in the image, utterly overlooking the real object, sulky, that the person is sitting on.

Delving into the architectures of these VLMs and how they infuse images into LLMs, we find that these VLMs deviate from the way human beings perform compositional reasoning from several perspectives. First, they feed one single image as a whole into LLMs and generate language descriptions based on the holistic image embedding. This is inconsistent with object-centric representations in vision, through which the whole image can be constituted by visual entities and more importantly, relationships between the entities. Second, these methods disregard the interaction between the sentence parts and the ingredients in the images. The generation of a new word by the LLM is not linked to a specific visual entity or relationship but is contingent on previous words and holistic image features instead. Although a series of works have been proposed to strengthen VLMs' compositional abilities (Doveh et al., 2023), they mainly probe the problem by proposing additional datasets. However, as stated by recent analysis on compositionality (Doveh et al., 2022), collecting specialized large-scale data to teach vision-language models the missing compositionality is impractical, as finding specialized text-image pairs for each kind and possible value of the visual entities and their relations is rather expensive. In this paper, we approach the essence of this problem from the perspective of model architecture, unveiling a compositional structure of LLM that can conduct step-by-step communication with visual components and relationships.

We propose CoVLM, which guides the LLM to explicitly compose visual entities and relationships among the text, and dynamically communicate with the detection network to achieve vision-language communicative decoding. Specifically, we devise a novel set of communication tokens for dynamic interaction and communication between the detection network and the LLM. Communication tokens are generated by the LLM, after the language tokens that denote visual entities or relationships. Upon the generation of communication tokens, a detection network is utilized to decode the regions relevant to the generated language sequence so far, and propose several bounding box proposals. The features of relevant regions are then fed back to LLM by communication tokens, conditioned on which the LLM decodes the subsequent tokens. The bottom-up vision-to-language and top-down language-to-vision communicative decoding are iteratively performed until all words and tokens are generated. The paradigm is shown on the right part of Figure 1.

We first evaluate our CoVLM on compositional reasoning tasks, including predicting the object entity given the subject and the relationship (ARO (Yuksekgonul et al., 2022)), matching the correct captions describing the relation between two images with similar entities (Cola, (Ray et al., 2023)), and human-object interaction detection (HICO-DET, (Chao et al., 2015)). We outperform baseline VLMs by a large margin (*e.g.*, $\sim 20\%$ in HICO-DET mAP, $\sim 14\%$ in Cola top-1 accuracy, and $\sim 3\%$ on ARO top-1 accuracy). We also achieve competitive results on vision-language tasks such as referring expression comprehension and visual question answering.

## 2 RELATED WORKS

### 2.1 VISION-LANGUAGE MODEL (VLM)

A proliferation of VLMs with remarkable commonsense reasoning abilities have been proposed recently. Among them, Flamingo (Alayrac et al., 2022) employs cross-attention and perceiver sampler to attend to visual contexts and enables visual context learning. BLIP2 (Li et al., 2023) uses a QFormer to attend to salient visual context for better language generation based on the visual context. LLaVA (Liu et al., 2023a) performs image-text alignment first and then conducts instruction finetuning. MiniGPT-4 (Zhu et al., 2023) aligns a frozen visual encoder with LLM using just one projection layer. mPLUG-Owl (Ye et al., 2023) also involves a two-stage method for aligning image and text. There are also recent papers that push VLMs to the 3D domain (Hong et al., 2023).

Recently, there has been a series of works that utilize LLMs for visual segmentation tasks. Specifically, VisionLLM (Wang et al., 2023) uses an LLM-based decoder which makes predictions about bounding boxes and polygons given language instructions. DetGPT (Pi et al., 2023) can interpret human instruction, reason about the visual scene with common sense knowledge, and finally output the objects of interest. GPT4RoI (Zhang et al., 2023) is capable of processing the user instructions that contain interleaved sequences of language and spatial information. LISA (Lai et al., 2023) proposes the embedding-as-mask paradigm to unlock the segmentation capability. However, the vision-language communication of these VLMs is one-way and one-time, merely using language instructions to generate segmentations or input segmented regions into the LLMs. KOSMOS-2 (Peng et al., 2023) infuses location tokens after visual entities into the language generation process. However, the communication is purely from the language system to the image for segmentation, while the grounded visual regions are not fed back to the language system. Furthermore, none of these VLMs tackle the relations or compositionality in the language inputs. In this paper, we propose CoVLM with a set of communication tokens for composing visual entities and relations and communicating between visual and language systems at each step.

### 2.2 COMPOSITIONALITY IN VISION AND LANGUAGE

Compositionality is a hallmark of human intelligence and plays an indispensable role in vision and language. Previous works exploring the compositionality in vision and language cover a variety of tasks such as visual question answering (Agrawal et al., 2017), generation (Liu et al., 2023b), retrieval (Saito et al., 2023), planning (Ajay et al., 2023) and so on. A set of datasets have been proposed for examining the compositionality of vision-language models (Hudson & Manning, 2019; Johnson et al., 2016; Agrawal et al., 2017; Krishna et al., 2017; Ma et al., 2023). Specifically, the Attribution, Relation, and Order (ARO) benchmark (Yuksekgonul et al., 2022) is a benchmark to systematically evaluate the ability of VLMs to understand different types of relationships, attributes, and order. Recently, VL-Checklist (Zhao et al., 2023) is a framework to evaluate VLM's abilities to recognize objects, attributes, and relations. Cola (Ray et al., 2023) analyzes VLMs' compositional ability in detail and proposes a text-to-image retrieval benchmark to compose objects with their relations. Evaluation of VLMs on these benchmarks and metrics shows current VLMs struggle with compositionality. Furthermore, a set of works find it particularly frustrating for VLMs when relationships are involved (Conwell & Ullman, 2022; Zhao et al., 2023). In this paper, we especially focus on relational compositionality, with the help of the aforementioned datasets and metrics.

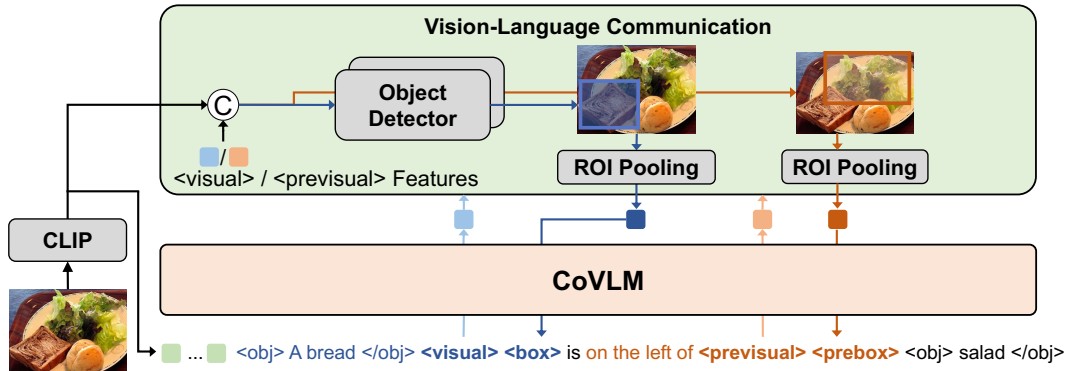

Figure 2: Overview of our CoVLM framework. Our vision module consists of a CLIP encoder to encode the image, and an object detector which takes in the image together with language inputs to generate relevant regions. For language modeling, we insert a set of communication tokens into the LLM, which can appear after a visual entity with a `<visual>` token or after a relationship with a `<previsual>` token. The last hidden layer of the LLM is then sent to the object detector to propose regions relevant to the language inputs so far. This is termed as top down language-to-vision communication. Next, in vision-to-language communication, the features of the proposed regions are fed back to LLM via `<box>` or `<prebox>` token for further language generation.

## 3  CoVLM

Most state-of-the-art Vision Language Models (*e.g.*, LLaVA (Liu et al., 2023a), Flamingo (Alayrac et al., 2022), BLIP-2 (Li et al., 2023)) take an image and text prompt as inputs and output a text sequence. Several recent VLMs (*e.g.*, VisionLLM (Wang et al., 2023), LISA (Lai et al., 2023), KOSMOS-2 (Peng et al., 2023)) enable a new ability to output segmentation masks based on the text input. Specifically, KOSMOS-2 generates a location token denoting a discretized bounding box for each visual entity to ground the visual entity to the image. However, the communication is purely from the LLM to the image for segmentation, while the grounded visual regions are not fed back to the LLM. In addition, the location tokens are generated after the visual entities, thus failing to assist in the process of generating word tokens based on grounded visual entities and relations. In short, the cut-off between the vision module and the LLM deprives previous VLMs of the crucial visual compositional ability. On the other hand, detection networks like Faster RCNN (Ren et al., 2016) can generate region proposals and classify the proposals, but can not interact with the language models.

In stark contrast to previous VLMs, our CoVLM stands out with its pioneering integration of detection networks into LLM to enable the seamless interaction between the vision module and the LLM and compositionality over visual entities and relations. As shown in Figure 2, we first devise a set of special communication tokens for flexibly switching between the visual detection module and the LLM. For LLM, we use a pre-trained Pythia model (Biderman et al., 2023) that can handle language tokens as inputs and outputs, as well as visual embeddings and special tokens which are mapped to the same embedding space as the language tokens, to constitute the LLM representations. The vision module consists of an image encoder that produces features to feed into the LLM and a detection network that proposes region proposals that are relevant to previous language inputs. Top-down language-to-vision communication is achieved by concatenating the last hidden state of the LLM-encoded features to the image embeddings and inputting them into the detection network, which proposes relevant regions conditioned on the LLM representations. Bottom-up vision-to-language communication extracts the features of the relevant regions and concatenates the features back into LLM for further language generation. In this way, the LLM is equipped with visual composionality. We give the details below.

### 3.1  VISION MODULE

Our vision module consists of two parts: an image encoder and a detection network.

**Image Encoder.** In this paper, we use the CLIP ViT-L model (Radford et al., 2021) for encoding the image. We use a linear mapping layer to map the image embeddings to the same embedding space as the Pythia language embedding space. We then append the image embeddings to the beginning of the language sequence.

**Detection Network.** Our detection network is similar to the YOLOX (Ge et al., 2021). The detection network takes as inputs two things: 1) the image embeddings of the whole image ($\mathcal{N} \times \mathcal{N} \times \mathcal{D}$, where $\mathcal{N}$ is the patch size and $\mathcal{D}$ is the embedding dim); 2) the last hidden state of the LLM so far ($1 \times \mathcal{D}$). The LLM embedding is expanded and concatenated to the same dim as the image embedding, yielding a final multi-modal embedding of size $\mathcal{N} \times \mathcal{N} \times 2\mathcal{D}$, and sent to the detection network. The detection network outputs $\mathcal{N} \times \mathcal{N} \times 4$ bounding boxes and $\mathcal{N} \times \mathcal{N}$ confidence scores. After non-maximum suppression, we keep a set of bounding boxes as regions of interest (ROIs). To extract the embeddings of one ROI, we extract the features of all patches that are covered by the ROI, and average pool to yield a box embedding of size $\mathcal{D}$. We choose the cropped image features of $m$ bounding boxes with top scores.

## 3.2 Language Models

We utilize the pre-trained Pythia model (Biderman et al., 2023) as the backbone of our LLM. In addition to language tokens, we also devise a set of special communication tokens to facilitate compositional vision-language modeling and communication, as is shown in Figure 2. We list the set of tokens below:

- **`<obj>, </obj>`**: these two tokens enclose a set of language tokens referring to a visual entity
- **`<visual>`**: this token is for switching to the vision module after a visual entity token $v_1$ is captured by LLM, so the vision module could attend to the visual entity
- **`<box>`**: this token receives the feedback from the vision module, concatenating the image features of detected $v_1$ back into the LLM
- **`<previsual>`**: this token is for switching to the vision module after a relation $r$ to a previous visual entity $v_1$ is detected (and before the visual entity $v_2$ that is in relation $r$ to $v_1$ is generated).
- **`<prebox>`**: this token switches back from the vision module after potential regions of $v_2$ are detected, and concatenating the features to better generate the language description of $v_2$.

The generation of communication tokens for visual entities and relations enables us to decompose the language sequences into smaller components, where each component connects to the vision module, thus improving compositionality.

## 3.3 Vision-Language Communication

The dynamic interaction and communication between the vision module and the language model can be iteratively performed through the special communication tokens introduced above.

**Top-Down Language-to-Vision Communication.** Top-down communication is achieved by first generating the `<visual>` token or `<previsual>` token. After the token is generated, we summarize the language information generated so far by taking the last hidden state of the LLM. This information gives the vision module a goal or task to attend to, just like the human visual system (Buschman & Miller, 2007). Contingent on the information so far, the vision module then uses the detection network to propose several ROIs, and extracts the features of these ROIs.

**Bottom-Up Vision-to-Language Communication.** Bottom-up communication is achieved by generating the `<box>` token or `<prebox>` token. The ROIs generated by the vision module are then fed back into the LLM to assist further language generation. For instance, if the `<prebox>` contains regions relevant to "a bread is on the left of", the LLM is then capable of absorbing this information and generating "salad".

## 3.4 Model Pre-training

**Pre-training data.** We create a large-scale grounded image-text dataset that consists of over 97M image-text pairs from the pre-training data of BLIP-2 (Li et al., 2023). The images are from a variety

| Model | ARO Accuracy | | Cola Accuracy | HICO-DET mAP | | |
|---|---|---|---|---|---|---|
| | Top-1 | Top-5 | Top-1 | Rare | Non-Rare | Full |
| *Vision-language Alignment Models* | | | | | | |
| CLIP (Radford et al., 2021) | 6.93 | 21.12 | 21.42 | - | - | - |
| FLAVA (Singh et al., 2022) | 4.59 | 12.76 | 24.76 | - | - | - |
| *Vision-language Generative Models* | | | | | | |
| OpenFlamingo3B (Awadalla et al., 2023) | 2.55 | 7.11 | 18.10 | - | - | - |
| BLIP (Li et al., 2022) | 29.78 | 54.18 | 41.43 | - | - | - |
| BLIP-2 ViT-L OPT$_{2.7B}$ (Li et al., 2023) | 29.73 | 54.91 | 35.71 | - | - | - |
| KOSMOS-2 (Peng et al., 2023) | 19.88 | 43.69 | 30.48 | 33.51 | 17.83 | 21.26 |
| **CoVLM 1.4B** | **32.46** | **55.70** | **44.29** | **50.82** | **35.47** | **39.00** |

Table 1: Compositional reasoning ability comparison with vision-language alignment models and generative models on three datasets. Visualization results are shown in the Appendix.

of datasets including COCO (Lin et al., 2014), CC3M (Sharma et al., 2018), CC12M (Changpinyo et al., 2021), Visual Genome (Krishna et al., 2017), SBU (Ordonez et al., 2011) and a subset of LAION400M (Schuhmann et al., 2021). Similar to KOSMOS-2 (Peng et al., 2023), we apply a grounding pipeline to the image-text pair to associate the text spans in the caption to their corresponding visual entities in the image. The pipeline consists of three steps: First, we use GroundingDINO (Liu et al., 2023c) to detect objects and their corresponding textual description. Inspired by KOSMOS-2 (Peng et al., 2023), we then apply spaCy (Honnibal et al., 2020) to expand the grounded words to grounded expressions to enrich their linguistic meaning. Finally, we insert communication tokens around the textual description to finalize the grounded data. More details about how we create the pre-training dataset can be found in the Appendix.

**Pre-training settings.** We trained two models: CoVLM 1.4B and 2.8B, which use Pythia-1.4B and Pythia-2.8B as the LLM respectively. Both of them use CLIP ViT-L/14 (Radford et al., 2021) as the image encoder. We load the huggingface checkpoint for these models and fully fine-tune the whole model during pre-training. More details can be found in the Appendix.

## 4 EXPERIMENTS

### 4.1 EVALUATION ON COMPOSITIONAL REASONING TASKS

We aim to probe the model's ability to reason about entities with detailed attributes in an image and also the relationships between two entities. Descriptions of the three datasets we are evaluating on below, the metric we use for evaluation, and the baselines we used are in the Appendix. All experiments are conducted in a zero-shot manner.

#### 4.1.1 ARO

**Setup.** For our model and other vision-language generative models, we feed the model with the image and text prompt "*entity_A relation*", considering the model output as predicted *entity_B*. For our model, we further insert a `<visual>` token after *entity_A* and a `<previsual>` token after *relation* to encourage the language model to better communicate with the visual branch. For vision-language alignment models, we use all 890 candidates *entity_B* to build 890 possible captions in the form of "*entity_A relation entity_B*" and pick the top-1/top-5 captions that have the highest similarity score with the image as the top-1/top-5 predictions.

**Results.** In Table 1, our model achieves superior performance, outperforming all other vision-language generative models and alignment models. It indicates that our model has a better ability to understand relations among visual entities in an image, and can better infer one visual entity using information of the presence of other visual entities and their relations. We also notice that the alignment models perform worse in this task. We hypothesize this is because the alignment models are trained using contrastive learning, which makes them behave like bag-of-words (Yuksekgonul et al., 2022). This makes them more easily to be misdirected by other objects in the image and produce the wrong prediction, instead of using the relationship to infer the correct one.

| Model | Zero-shot | mAP | | |
| | | Rare | Non-Rare | Full |
|---|---|---|---|---|
| InteractNet (Gkioxari et al., 2018) | ✗ | 7.16 | 10.77 | 9.94 |
| CDN (Zhang et al., 2021) | ✗ | 27.39 | 32.64 | 31.44 |
| GEN-VLKT (Liao et al., 2022) | ✗ | 29.25 | 35.10 | 33.75 |
| RLIPv1-ParSe (Yuan et al., 2022) | ✗ | 26.85 | 34.63 | 32.84 |
| RLIPv2-ParSeDA (Yuan et al., 2023) | ✗ | 43.23 | **45.64** | **45.09** |
| RLIPv1-ParSe (Yuan et al., 2022) | ✓ | 15.08 | 15.50 | 15.40 |
| RLIPv2-ParSeDA (Yuan et al., 2023) | ✓ | 27.97 | 21.90 | 23.29 |
| CoVLM 1.4B | ✓ | **50.82** | 35.47 | 39.00 |

Table 3: Comparison with task-specific methods on HICO-DET.

### 4.1.2 COLA

**Setup.** For our model and other vision-language generative models, we calculate the perplexity score between a caption and all candidate images and choose the image with lower perplexity as the prediction. Specifically, we feed the model with one image and a caption in the form of "*entity_a relation entity_b*". We calculate the average perplexity of the overall text output. Notably, for our model, we will insert a `<visual>` and `<previsual>` tokens after *entity_a* and *relation*, respectively to encourage vision-language communication. For vision-language alignment models, we directly report the results from Cola (Ray et al., 2023).

**Results.**

In Table 1, our CoVLM significantly outperforms both alignment and generative methods by a large margin. We attribute the performance to the `<previsual>` token which helps to retrieve the visual information of the *entity_b* for better describing its detailed attributes in text form, thus leading to lower complexity for the ground-truth caption. Also, the `<visual>` token helps to better localize *entity_a*, allowing the model to better localize the area of *entity_b* according to *relation*. In Table 2, we also com-

| Model | Acc. |
|---|---|
| CLIP + MM-Pred (Ray et al., 2023) | 41.42 |
| CLIP + MM-Adapter (Ray et al., 2023) | 40.95 |
| FLAVA + MM-Pred (Ray et al., 2023) | 39.04 |
| FLAVA + MM-Adapter (Ray et al., 2023) | 40.47 |
| CoVLM 1.4B | **44.29** |

Table 2: Comparisons with task-specific supervised learning methods on Cola.

pare our zero-shot results with task-specific methods proposed in Ray et al. (2023) which fine-tunes CLIP (Radford et al., 2021) and FLAVA (Singh et al., 2022) on the training data of Cola. Our method still achieves the best performance, demonstrating the superiority of generalization and robustness of our model.

### 4.1.3 HICO-DET

**Setup.** For our method and other generative models, we predict HOI (*subject, verb, object*) in two steps: a) recognizing the interaction categories represented by *verb*, and b) localizing the *subject* and *object*. To determine the existence of interaction, we manually build positive and negative phases with *verb*, *i.e.,* "*the person is verb*" and "*the person is not verb*" and calculate their perplexities using the generative model. If the perplexity of the positive phase is lower than the negative one, we consider this *verb* exists in the image. For all detected *verb*, we feed the model with the image and text prompt "*the person is verb*" to predict *object* and the location of the person and object. Notably, for our CoVLM, we use the inserted `<visual>` and `<previsual>` tokens after *the person* and *verb* to predict the locations. Since the output of alignment methods does not contain the object location, we ignore these methods on the HICO-DET dataset.

**Results.** Table 1 presents the zero-shot results on the HICO-DET test set. Our model significantly outperforms KOSMOS-2 (Peng et al., 2023). We attribute the performance improvement to the `<previsual>` token that forces the model to communicate with the input image to localize the area of *object* and predict the text of *object*. In comparison, KOSMOS-2 only feeds the image

| Model | RefCOCOg | | RefCOCO+ | | | RefCOCO | | |
|---|---|---|---|---|---|---|---|---|
| | val | test | val | testA | testB | val | testA | testB |
| GRILL (Jin et al., 2023) | - | 47.50 | - | - | - | - | - | - |
| ReCLIP (Subramanian et al., 2022) | 59.33 | 59.01 | 47.87 | 50.10 | 45.10 | 45.78 | 46.10 | 47.07 |
| KOSMOS-2 (Peng et al., 2023) | 60.57 | 61.65 | 45.48 | 50.73 | 42.24 | **52.32** | **57.42** | **47.26** |
| **CoVLM 1.4B** | 60.87 | 61.91 | 47.62 | 50.93 | 44.16 | 48.19 | 53.17 | 43.18 |
| **CoVLM 2.8B** | **61.23** | **62.33** | **48.87** | **52.51** | **44.71** | 49.32 | 53.67 | 44.49 |

Table 4: Comparison of referring expression comprehension on three datasets.

information at the beginning of the input, and thus the model may suffer from language prior to predicting a wrong *object*. In Table 3, we also compare our zero-shot results with the task-specific supervised learning methods. Our model achieves comparable results on the Non-Rare and Full sets. Notably, our zero-shot result exceeds all supervised learning methods in the Rare set, demonstrating the generalization ability of our model.

## 4.2 EVALUATION ON VISION-LANGUAGE TASKS

In addition to compositional reasoning tasks, we also evaluate object localization and complex vision-language understanding abilities of our model. All experiments are conducted in a zero-shot manner.

### 4.2.1 REFERRING EXPRESSION COMPREHENSION

**Setup.** This task requires a model to locate the bounding box of an object specified by language description. We follow KOSMOS-2 (Peng et al., 2023) to use three well-established benchmarks namely RefCOCOg (Mao et al., 2016), RefCOCO+ (Yu et al., 2016) and RefCOCO (Yu et al., 2016). All these datasets use images from COCO (Lin et al., 2014). Both RefCOCO and RefCOCO+ are annotated by a two-player game, where the spatial relations are excluded in RefCOCO+. The RefCOCOg contains longer referring expressions and spatial relations.

For our CoVLM, we feed the model with `<obj>expression</obj><visual>` and the `<visual>` will generate multiple bounding boxes with their confidence scores. Instead of choosing the bounding box with the highest score as a prediction, we use `<previsual>` to further measure the alignment between the box and expression. Specifically, we select the bounding box with the highest confidence score and subsequently choose additional bounding boxes from the remaining set whose confidence scores exceed a predefined threshold (0.5 times the highest score in our case) as candidates. For each candidate, we feed the model with `<previsual><prebox><obj>expression</obj>` and put the image feature of the bounding box region into `<prebox>`. Then we calculate the perplexity of the *expression* for this bounding box candidate and choose the one with the lowest perplexity as our final prediction.

**Results.** In Table 4, our CoVLM 2.8B variant performs the best on both val and test sets of RefCOCOg and RefCOCO+, demonstrating its superior localization ability. On RefCOCO, we achieve comparable performance with KOSMOS-2. Note that KOSMOS-2 has been instruction fine-tuned on the data with the same form as the referring expression task, while our CoVLM is directly transferred to this new task without any form of instruction fine-tuning.

### 4.2.2 VISUAL QUESTION ANSWERING

**Setup.** This task requires a model to answer questions about an image. Following BLIP-2 (Li et al., 2023), we evaluate on VQAv2 (Goyal et al., 2017) dataset and report the zero-shot accuracy on the test-dev set. We use the prompt "*Question: {question} Short Answer:*".

**Results.** In Figure 3, our CoVLM 2.8B model achieves better performance compared with MetaLM (Hao et al., 2022), VLKD (Dai et al., 2022), OpenFlamingo3B (Awadalla et al., 2023), and KOSMOS-2 (Peng et al., 2023), and has a small margin compared with Flamingo3B (Alayrac et al., 2022) and BLIP-2 ViT-L OPT$_{2.7B}$ Li et al. (2023). We hypothesize the accuracy margin may stem from the generative model generating diverse answers that align conceptually with ground

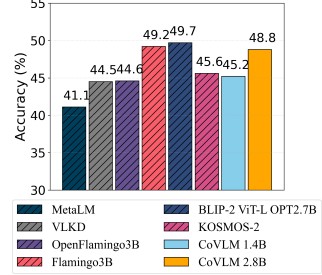

Figure 3: VQAv2 test-dev results.

| Setting | ⟨previsual⟩ | ⟨visual⟩ | ⟨prebox⟩ | ⟨box⟩ | ARO Top-1 | Cola |
|---------|-------------|----------|----------|-------|-----------|------|
| 1 | ✗ | ✗ | ✗ | ✗ | 29.60 | 38.10 |
| 2 | ✗ | ✓ | ✗ | ✓ | 30.08 | 34.76 |
| 3 | ✓ | ✓ | ✗ | ✗ | 32.26 | 36.19 |
| 4 | ✓ | ✓ | ✓ | ✓ | **32.46** | **44.29** |

Table 5: Ablation study of different sets of communication tokens.

truth, yet may not exactly match the annotation, affecting the evaluation. To get a better insight into how well our model performs on VQAv2, we conduct a round of human evaluation for our CoVLM 2.8B model and BLIP-2 ViT-L OPT$_{2.7B}$ model on a randomly selected subset with 1000 samples. The human evaluation accuracy for our model is $57.11$ while the accuracy for BLIP-2 is $56.62$, suggesting that the performance gap between our model and BLIP-2 is negligible.

### 4.3 ABLATION STUDY

Compared to other VLMs, our CoVLM introduces a novel set of communication tokens to improve the compositional reasoning ability. To evaluate the effectiveness of each type of communication token and the effectiveness of the design of bidirectional communication of the visual module and language module, we create the following four settings: 1) No communication token at all. 2) No `<previsual>` and `<prebox>`, so there are no communication tokens to compose a relationship. 3) No `<prebox>` and `<box>`, so the communication only happens from the language module to the vision module. 4) All communication tokens are presented. For each setting, we pre-train our CoVLM 1.4B model under the same setting and evaluate on ARO and Cola benchmarks. Table 5 shows the evaluation results for these four settings. We summarize the insights as follows:

1. With no communication tokens, our model's performance on compositional reasoning tasks is very similar to BLIP-2. This is reasonable because we share the same pre-training data with BLIP-2, thus also inheriting a similar performance on downstream tasks.

2. Generate `<visual>`/`<box>` after the object does not help compositional reasoning. It is reasonable because adding extra information after the object description won't help the generation of that object description itself as this is an auto-regressive generation process.

3. Not putting `<prebox>`/`<box>` back into the generated sequence will hurt the compositional reasoning ability for complex object description. We can find this insight in the last two rows of the ablation study table. ARO Top-1 accuracy does not hurt much if we do not put `<prebox>`/`<box>` back, while the performance for Cola will drop significantly. This is because in ARO, the predicted object is usually a simple phrase without any attribute, such as "horse" and "car". During training, the model can learn to bind this kind of simple concept with the `<previsual>`/`<visual>` tokens, so merely generating `<previsual>`/`<visual>` is adequate for enhancing the model's compositional reasoning ability on ARO benchmark. However, the object description in Cola is much more complex, such as "yellow vehicle" and "standing man", which requires a close inspection of the visual feature of the object. In this case, `<prebox>`/`<box>` tokens which contain more fine-grained information about the visual entity can play an important role in assisting LLM to focus on these complex objects and generate more faithful and related tokens thereafter.

## 5 CONCLUSION

In this paper we propose CoVLM, which can guide the LLM to explicitly compose visual entities and relationships among the text, and dynamically communicate with the detection networks to achieve vision-language communicative decoding. We outperform previous VLMs by a large margin on compositional reasoning benchmarks (*e.g.*, $\sim 20\%$ in HICO-DET mAP, $\sim 14\%$ in Cola top-1 accuracy, and $\sim 3\%$ in ARO top-1 accuracy). We also achieve competitive performances on referring expression comprehension and visual question answering. However, we do not yet address object-attribute and spatial event compositionally much, which are crucial future directions.

## ACKNOWLEDGEMENT

We are grateful to the anonymous reviewers for their valuable feedback. This work was partially funded by grants from Google, Amazon, and Adobe. We also extend our thanks to AiMOS for supplying the computational resources necessary for this project.

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

# A APPENDIX

## A.1 PSEUDO ALGORITHM FOR COMMUNICATIVE DECODING

---
**Algorithm 1** Communicative Decoding
---
**Input:** 224x224 RGB image $I_{raw}$ and text prompt $P_{raw}$
**Output:** VLM generated tokens $S$
1:  $I \leftarrow$ vision_encoder($I_{raw}$)
2:  $P \leftarrow$ text_embedding($P_{raw}$)
3:  $S \leftarrow$ empty list
4:  $S \leftarrow$ Append($S, I$)
5:  $S \leftarrow$ Append($S, P$)
6: **repeat**
7:     $X \leftarrow$ Auto-regressively predicted next token of $S$
8:     $S \leftarrow$ Append($S, X$)
9:     **if** $X$ is `<previsual>` **then**
10:       $H \leftarrow$ last hidden state of `<previsual>`
11:       $ROIs \leftarrow$ detector($I, H$)
12:       **for** each $ROI \in ROIs$ **do**
13:         $I_{ROI} \leftarrow$ Crop($I, ROI$)
14:         `<prebox>` $\leftarrow$ MeanPooling($I_{ROI}$)
15:         $S \leftarrow$ Append($S,$ `<prebox>`)
16:       **end for**
17:     **else if** $X$ is `<visual>` **then**
18:       $S \leftarrow$ Delete the latest `<previsual>` and all its corresponding `<prebox>` in $S$
19:       $H \leftarrow$ last hidden state of `<visual>`
20:       $ROIs \leftarrow$ detector($I, H$)
21:       **for** each $ROI \in ROIs$ **do**
22:         $I_{ROI} \leftarrow$ Crop($I, ROI$)
23:         `<box>` $\leftarrow$ MeanPooling($I_{ROI}$)
24:         $S \leftarrow$ Append($S,$ `<box>`)
25:       **end for**
26:     **end if**
27: **until** $X$ is `EOS` token
---

Algorithm 1 describes how our communicative decoding works. Our model takes an RGB image with a resolution of 224x224 and a text prompt as input. The RGB image will first be encoded to an image embedding by the vision encoder and the text prompt will be encoded by the LLM's text embedding. Then, we put the image embedding in front of the encoded prompt to form a multimodal prompt, and we feed this multimodal prompt into LLM for auto-regressive generation.

The generation process are mostly the same as the LLM's original auto-regressive next token prediction process, except that we include an iterative and automatic insertion of communication tokens to facilitate the compositional reasoning ability. `<previsual>` is automatically generated by LLM via next token prediction, and arbitrary number of ROIs can be detected through the object detector module. We set the confidence score threshold to be 0.05 and retain all ROIs that has a confidence score higher than 0.05. For each ROI, we crop that region from image embedding, and perform a mean pooling to get a token, namely `<prebox>`, that contains the visual information of the ROI. All `<prebox>`s are automatically inserted into the token sequences so that the future generation process can use the information of these tokens. The generation of `<visual>` and `<box>` are similar to `<previsual>` and `<prebox>`. In order to prevent the localization information leak induced by the latest `<previsual>` and `<prebox>`, we delete the latest `<previsual>` and `<prebox>` before we use the last hidden state of the new `<visual>` to detect objects.

## A.2 PRE-TRAINING DATASET

Similar to KOSMOS-2 (Peng et al., 2023), we adopt a pipeline making use of out-of-the-box open vocabulary detector to create our grounded pre-training dataset. It conssts of three steps:

**Step-1: Generating bounding-box-word pairs.** We use GroundingDINO (Liu et al., 2023c) to detect objects in the image and link the bounding box of the object to words in the text. We keep bounding boxes whose highest similarities are higher than $0.35$ and extract the words whose similarities are higher than $0.25$ as the words that correspond to a bounding box. Non-maximum suppression algorithm is applied to eliminate bounding boxes that have a high overlap with other bounding boxes linked to the same word.

**Step-2: Expanding grounded words to grounded expressions.** In practice, we observe that GroundingDINO often fail to link the whole referring expressions to an object in the image. For example, for the expression "man with a hat on his head", GroundingDINO will only link "man" to the person in the image, but not the whole expression. This will limit the model's ability to understand complicated expressions. Inspired by KOSMOS-2 (Peng et al., 2023), we apply spaCy (Honnibal et al., 2020) to obtain each word's dependency relation in the sentence, and expand a grounded word to a grounded expression by recursively traversing the dependency tree of that word and concatenate eligible children words based on the linguistic rules.

**Step-3: Assigning bounding boxes to the special communication tokens.** Given the expressions and their associated bounding boxes in a grounded image-text pair, we can now insert the special communication tokens into the text and assign the bounding boxes to them. We follow KOSMOS-2 (Peng et al., 2023) to enclose the expression in `<obj>`/`</obj>` pair, and then add communication tokens around them. For a given expression with a single bounding box, the resulted input sequence for that expression is either in the form of "`<obj>`expression`</obj><visual><box>`" or "`<previsual><prebox><obj>`expression`</obj>`" depending on the position of the expression in the sentence. If an expression is associated with multiple bounding boxes, we add multiple `<prebox>` or `<box>`. If it is the first expression in the sentence, we use the form with a trailing `<visual>` token. Otherwise, we randomly select one from these two available forms.

### A.3 Pre-training Details

Apart from the grounded image-text pair dataset we created, we also use The Pile (Gao et al., 2020) as part of our pre-training dataset. The total pre-training loss consists of the language modeling loss and the detection loss, with a loss weight of $0.025$ for the detection loss. We pre-train for 20k steps and use a batch size of 2,304 for grounded image-text data and a batch size of 2,304 for The Pile data. AdamW (Loshchilov & Hutter, 2017) optimizer is employed with a learning rate of $1.0e^{-4}$ and $\beta = (0.9, 0.999)$. We do not apply weight decay to the weights of LLM and CLIP, but apply a weight decay of $0.05$ to the detection network.

### A.4 Compositional Evaluation Dataset

**Datasets and Metrics.** We conduct experiments on ARO (Yuksekgonul et al., 2022), Cola (Ray et al., 2023), and HICO-DET (Chao et al., 2015) datasets.

- **ARO** contains 23,937 testing images. One entity pair in an image is annotated as a tuple (*entity_A*, *relation*, *entity_B*). Given *entity_A* and *relation*, the model is required to predict *entity_B* out of all candidate objects. We use top-1 and top-5 accuracy as evaluation metrics.

- **Cola** contains 420 testing images, where each image is paired with a caption describing the relation between two entities similar to ARO. The entity in Cola is described with more attribute details (*e.g.*, texture, color, and size) which require the model to perform more fine-grained object recognition. We conduct experiments on the multi-object part of this dataset, where two images with similar entities and their corresponding captions are paired as a testing sample. The model is required to correctly match the corresponding caption for both two images. We report the accuracy following Cola (Ray et al., 2023).

- **HICO-DET** contains 9,658 testing images, with 600 HOI categories constructed by 80 object categories and 117 verb classes. Each instance of human-object interaction is represented as a triplet, denoted as (*subject*, *verb*, *object*), accompanied by their respective bounding boxes. The model is required to not only recognize the HOI categories but also localize the *subject* and *object*. Following Chao et al. (2015), we report the mean AP (mAP) on 3 splits, namely a) Rare: 138 HOI categories with less than 10 training instances, b) Non-Rare: the remaining 462 HOI categories, and c) Full: all 600 HOI categories.

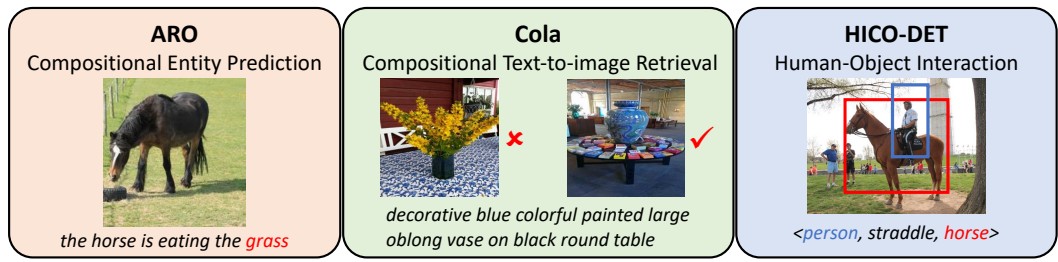

Figure 4: Explanation of three evaluated compositional reasoning tasks.

**Baselines.** We mainly compare our CoVLM with two types of methods, namely the vision-language alignment model (*i.e.*, CLIP (Radford et al., 2021), FLAVA (Singh et al., 2022)) and the vision-language generative model (*i.e.*, OpenFlamingo (Awadalla et al., 2023), BLIP-2 (Li et al., 2023) and KOSMOS-2 (Peng et al., 2023)). The vision-language alignment models learn a vision encoder and a language encoder, which pull close the features of paired image and text while pushing away the unpaired one. The vision-language generative model takes as input the image and text and auto-regressively generates a sequence of text.

## A.5 VISUALIZATION OF COMPOSITIONAL REASONING RESULTS

We show the visualization results for our model, BLIP-2 and KOSMOS-2 on the three compositional reasoning tasks.

Figure 5 shows the qualitative results for ARO. Compared to BLIP-2 and KOSMOS-2, we can rank the ground truth object higher thanks to the <previsual> token to localize the object.

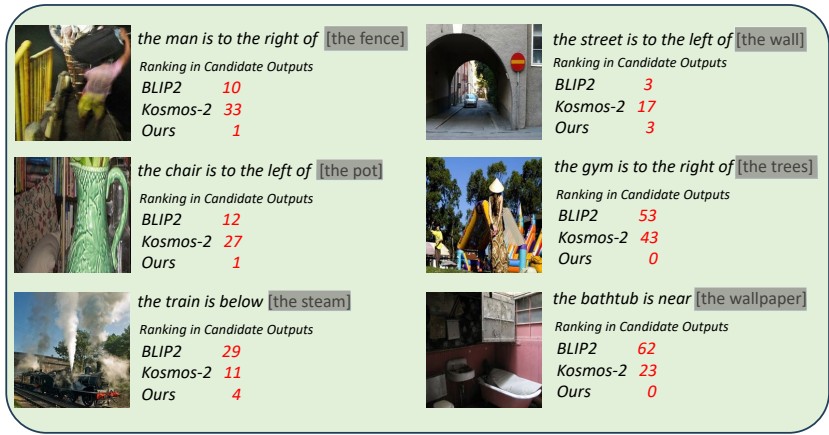

Figure 5: Qualitative results on ARO.

Figure 6 shows the results for Cola. The presence of the <prebox> token can encode the visual feature of the ROI, helping the model better understand the attribute of the objects due to its zoom-in effect. Also, relationship can be explicitly used by <previsual> token to better infer the visual entity.

Figure 7 shows the comparison of our model and KOSMOS-2 on the HICO-DET task. In general our model gives more accurate bounding box for object localization. Also, when there are two identical objects in the image, *i.e.*, the bottom-right example in Figure 7, our model can take advantage of the <previsual> token which can guide our model to pay attention to the region of the correct object (the bench that the person is lying on), instead of the wrong object (the bench that behind the person).

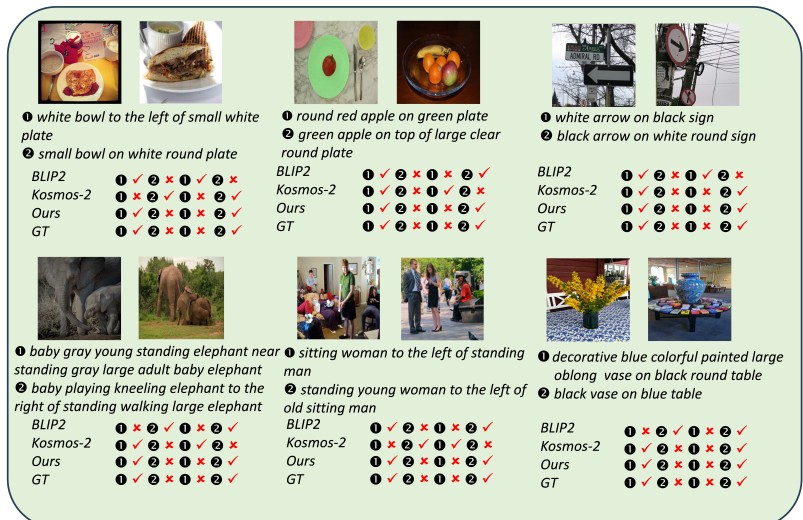

Figure 6: Qualitative results on Cola.

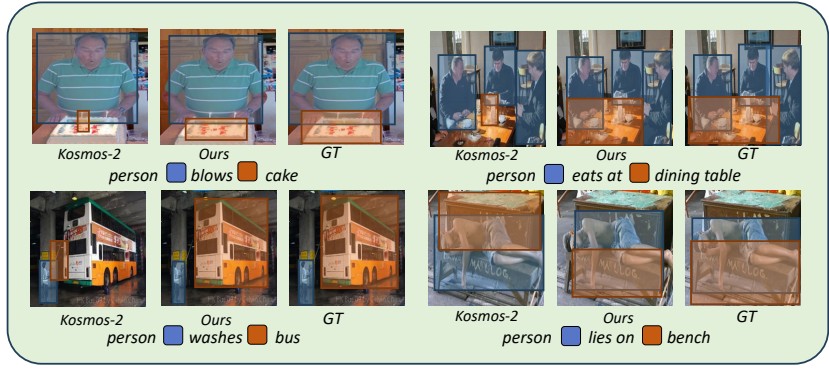

Figure 7: Qualitative results on HICO-DET.

Figure 8 illustrates the performance consistency of our model across diverse and ambiguous scenarios. Specifically, in instances where the expression inputs exhibit ambiguity — such as having multiple boxes corresponding to the same text — our model generates all potential bounding boxes with scores surpassing the threshold.

Figure 9 depicts the functioning of the model in the image captioning task. Notably, when the model produces nouns with modifiers, such as "an old man", "A red fire hydrant" and "a glass of champagne", it considers them as integral units and automatically generates "<obj>an old man</obj>", "<obj>A red fire hydrant</obj>", "<obj>a glass of champagne</obj>", instead of parsing the objects individually, for instance, "an old <obj>man</obj>" and "A red <obj>fire hydrant</obj>".

Figure 10 delineates instances of model failures. Various texts were input under the same picture to obtain the output. As depicted in (a), our model proficiently recognizes normal-sized objects. However, challenges arise when dealing with minuscule or blurred objects, such as the "spoon" depicted in the picture, making it difficult for the model to yield the desired results. Moreover, as illustrated in (b), the model successfully identifies aligned objects of the same kind in the picture. While our model adeptly produces corresponding bounding boxes when multiple objects of the same kind are neatly arranged, the scenario becomes more challenging in crowded scenes or when the object arrangement is irregular, often resulting in the model's inability to output all the bounding boxes.

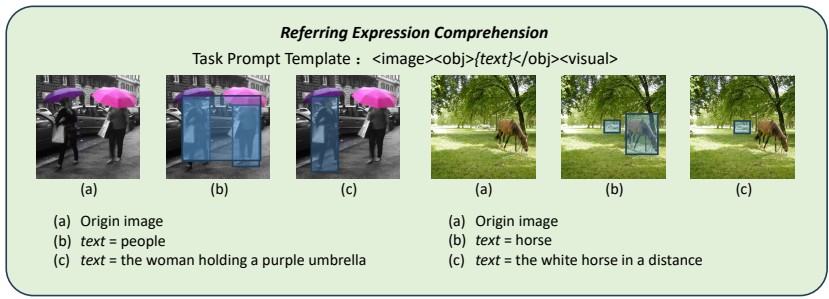

Figure 8: Qualitative results on Referring Expression Comprehension with various boxes output.

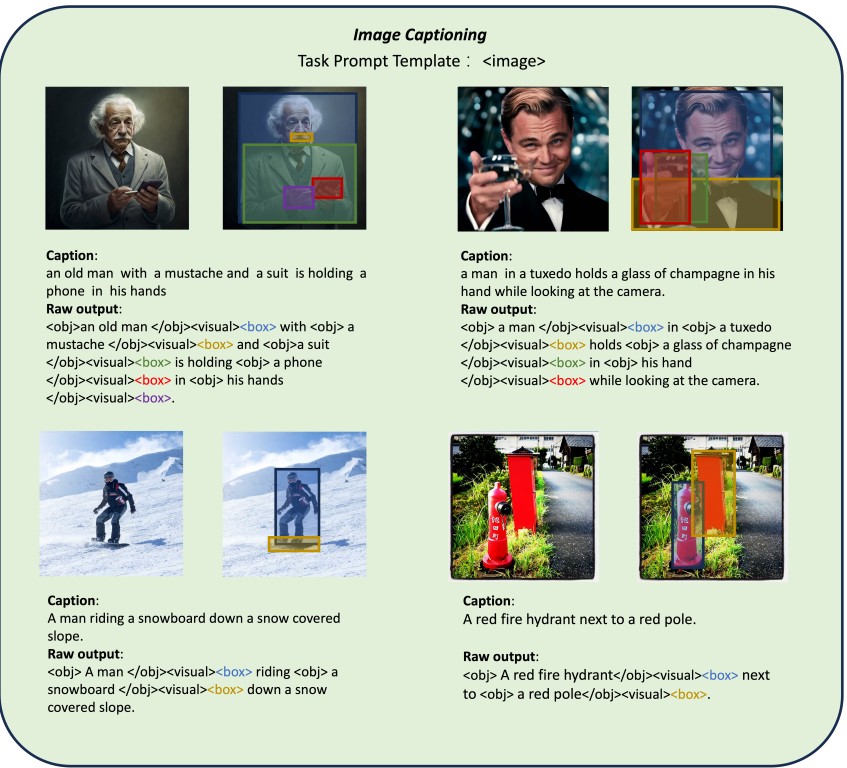

Figure 9: Qualitative results on Image Caption.

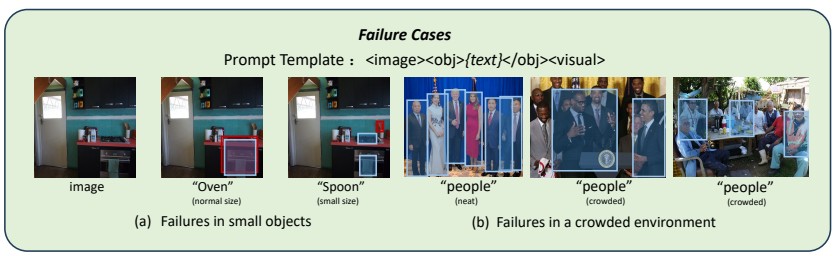

Figure 10: Qualitative results on failure cases.

