# OpenReview forum: "CoVLM: Composing Visual Entities and Relationships in Large Language Models Via Communicative Decoding"
_ICLR.cc/2024/Conference — ICLR 2024 poster_

### Official Review · Reviewer_fA94 · 2023-10-22

**Soundness:** 2 fair
**Presentation:** 4 excellent
**Contribution:** 3 good
**Rating:** 8
**Confidence:** 4

**Summary:**

Current vision-language models (VLMs) lack this ability due to their "bag-of-words" approach and inability to accurately represent visual entities and their relationships. The proposed Compositional VLM addresses this by guiding the model to explicitly compose visual entities and relationships and enabling dynamic communication between the visual detection system and the language system. This is achieved through the introduction of communication tokens, which guide the detection network to propose relevant visual regions based on the generated sentence. These regions are then integrated into the language generation process. This iterative communication between vision and language continues until a complete sentence is generated. This approach effectively bridges the gap between visual perception and language models, significantly outperforming other VLMs in compositional reasoning benchmarks. It also performs well in traditional vision-language tasks like referring expression comprehension and visual question answering.

**Strengths:**

1. **Dataset Collection**: The paper collect new object level bounding box to image-text pairs ensures that the data is not just vast but also well-curated, enhancing the model's training and performance.

2. **Reasonable Model Design**: The introduction of the Compositional VLM, with the token design framwork and the interation between visual detector LLM, displays a systematic approach towards bridging the gap between visual entities and their textual descriptions.

3. **Comprehensive Experiments**: With the grounded image-text pairs, the paper offers a detailed experimental setup, validating the model across various compositional reasoning benchmarks.

3. **Strong Performance**: The Compositional VLM showcases impressive results, especially when compared to previous vision-language models.

**Weaknesses:**

1. **Fundamental Oversimplification of Compositionality:** The Compositional VLM framework, though innovative, may not truly capture the essence of disentangling objects and their relationships within images. Instead of delving deep into the inherent complexities of this challenge, the method leans towards gathering object-text-image paired datasets and reinforcing their connections. This approach, while seemingly effective, might only be a surface-level solution rather than addressing the root of the compositional problem.

2. **Scalability Concerns**: The model's heavy reliance on precise associations between text captions and visual entities raises questions about its scalability. Can it consistently perform well in diverse or ambiguous scenarios, or will it be constrained by the specificity of its training data?

3. **Rigid Token Implementation**: The manual nature of token positioning (with <obj> around the object and <box> after them), suggests a lack of flexibility in the model. This rigidity could hamper the model's adaptability, especially when faced with varied or unforeseen testing scenarios. For example, if we have a "a man between two dog", how to deal with the <box> for multiple instance of dog.

4. **Operational Inefficiency**: The necessity for pre-parsing sentences and manually inserting tokens, even during testing, indicates potential operational bottlenecks. This could impede real-time applications and demands additional preprocessing steps, detracting from the model's overall efficiency.

**Questions:**

1. When integrating the object detection model within the auto-regressive language model, what would be the time complexity for each individual inference? Would it be very slow?

2. Given that the proposed Compositional VLM necessitates text parsing even during inference, how resilient is the model to inaccuracies or ambiguities in token placement? For instance, the phrase "a black dog" could be parsed as "a <obj>black dog<\obj>" or "a black <obj>dog<\obj>". How does the model handle such variations?

**Details Of Ethics Concerns:**

New collected dataset, may require ethic review.

---

> ### Author Response · Authors · 2023-11-16
> **Response to Reviewer fA94 (part1)**
>
> We thank you for your time and valuable comments. Below we answer the main concerns raised in the review and would be happy to provide further clarification if suitable.
>
> > **Q1. What would be the time complexity for each individual inference? Would it be very slow?**
>
> In our implementation, we use the YOLOX detection head as our object detection model, which is very lightweight compared to the auto-regressive language model and CLIP vision encoder. The number of parameters for the object detection model is negligible compared to the whole model.
>
> |  | #Params | Latency |
> |:---:|:---:|:---:|
> | Object detector | 3.4M | 2.2 ms |
> | Vision encoder + LLM | 1,717.8M | 103.9 ms |
> | Percentage | 0.2% | 2.1% |
>
>
>
>
> We also measure the latency for the object detector and the vision encoder + LLM, separately. From the table above, the object detector takes only 2.1% of the total latency. Since the object detection model is not activated every time during training and inference, it will take less than 2.1\% time when we generate a sequence of tokens.
>
> > **Q2. How resilient the model is to inaccuracies or ambiguities in token placement.**
>
> Our model is resilient to inaccuracies or ambiguities in token placement. We evaluate the token placement accuracy and our model's resilience to inaccurate token placement on RefCOCOg/RefCOCO+/RefCOCO datasets.
> * Token placement accuracy measures how accurate *\<obj\>* and *\</obj\>* is placed compared to the ground truth. We handcraft several templates to contain the referring expression into a sentence. For example, one template is "I can see {expression} in the image". If both *\<obj\>* and *\</obj\>* are placed correctly, we count it as a true positive in overall accuracy. The placement accuracy for *\<obj\>* and *\</obj\>* are also separately evaluated.
> * Localization accuracy measures the correctness of the predicted bounding box using either ground truth placement or predicted placement for *\<obj\>* and *\</obj\>*.
>
> |  | RefCOCOg val | RefCOCO+ val | RefCOCO val |
> |---|:---:|:---:|:---:|
> | **Token Placement Accuracy** |  |  |  |
> | Overall | 13.21 | 47.43 | 44.30 |
> | <obj> | 99.84 | 99.84 | 99.85 |
> | </obj> | 13.21 | 47.43 | 44.30 |
> | **Localization Accuracy** |  |  |  |
> | Predicted Placement | 58.74 | 45.77 | 45.71 |
> | Ground Truth Placement | 60.87 (+2.13) | 47.62 (+1.85) | 48.19 (+2.48) |
>
>
> For token placement accuracy, the results show that the overall accuracy is not bad except on the RefCOCOg dataset, which mostly consists of long expressions. Therefore, it is harder for our model to correctly identify the whole expression. We further analyze the placement accuracy for *\<obj\>* and *\</obj\>* separately, and we can find that our model can generate *\<obj\>* in the right place in most cases, while the generation of *\</obj\>* is not always correct. We empirically find that our model often fails in cases such as "an adult giraffe scratching its back with its horn". It will be parsed as "*<obj>*an adult giraffe scratching its back*</obj>* with its horn" or "*<obj>*an adult giraffe*</obj>* scratching its back with its horn".
>
> From the results of localization accuracy, we can see that the accuracy of using predicted token placement and the accuracy of using ground truth token placement is close. It indicates that our model is robust to inaccurate or ambiguous token placement.
>
> > **Q3. Fundamental oversimplification of compositionality.**
>
> Bringing compositionality to VLM is a challenging problem, and we are the very first step to handle this challenge. The good results in representative compositional reasoning benchmarks show the effectiveness of our method in improving the compositional reasoning ability of VLMs, and we acknowledge that there are some limitations in our method such as not coping much with object-attribute compositionality and spatial event compositionality, which could be crucial future directions.
>
> > **Q4. Rigid token implementation. If we have a "a man between two dog", how to deal with the <box> for multiple instances of dog?**
>
> Our model can deal with the situation that there are multiple objects in an image, thanks to our detector (i.e., a YOLO-style model) can localize them simultaneously.
>
> Taking the text "*a man between two dogs*" as an example, a *<visual>* token is automatically generated after the word "dogs". Then, the object detector takes this token as input and outputs two boxes, each for one dog. All the features of predicted boxes will be sent back to LLMs, using one *<box>* for each detected box. Thus, the updated text would be "*a man between two \<obj\>dog\</obj\><visual><box><box>*". This gives our model the maximum flexibility to handle various scenarios.

---

> > ### Author Response · Authors · 2023-11-16
> > **Response to Reviewer fA94 (part2)**
> >
> > > **Q5. The necessity for pre-parsing sentences and manually inserting tokens, even during testing, indicates potential operational bottlenecks.**
> >
> > Sorry for the confusion. Actually, all communication tokens are **automatically** generated by LLM. During the auto-regressive generation, users do not need to pre-parsing sentences nor manually insert communication tokens. We have provided more details about communication token generation in G1.

---

> > > ### Comment · Reviewer_fA94 · 2023-11-20
> > > **Thanks for the response**
> > >
> > > Thank you for the detailed clarification, especially regarding the experiments with token replacement. This addresses my concerns about the accuracy of token placement effectively. However, I believe the approach might be overly simplistic. The claim that this is the 'very first step to handle this challenge' seems overstated, since the compositionality in VLM is prevailing, and has been repeated in brought up in the field.

---

> > > > ### Author Response · Authors · 2023-11-21
> > > > **Followup (part1)**
> > > >
> > > > > I believe the approach might be overly simplistic. The claim that this is the 'very first step to handle this challenge' seems overstated, since the compositionality in VLM is prevailing, and has been repeated in brought up in the field.
> > > >
> > > > Thank you for your comment. We agree with the Reviewer that compositionality is an important topic and worthy of paying great attention to, and that's the reason why there are a lot of related works focusing on this topic [a,b,c,d]. However, our method is significantly different from [a,b,c,d] in the aspect of model architecture and training data.
> > > >
> > > > 1. For model architecture, thanks to the auto-regressive generative LLM, our model can achieve **free-form generation**, so it is more flexible and can fit into tasks with any output format, bringing great **generalization** to various tasks. The integrated detection network in our method can help **visualize and interpret** the compositional reasoning process, highlighting which region our model is focused on during the generation process.
> > > > 2. For training data, instead of using rule-based templates or synthetic data, we use real-world images and captions from a collection of datasets and apply an open-vocabulary detector to automatically label the data. Therefore, our training data is **diverse** and **realistic** with **fewer hallucinations**, making our model possible to apply to diverse datasets and tasks.
> > > >
> > > > We summarize the differences between our method and previous works [a,b,c,d] in the following table, and compare our method with [a,b,c,d] in detail:
> > > >
> > > >
> > > > |  | MosaiCLIP [a] | SyViC [b] | DAC [c] | TSVLC [d] | Ours |
> > > > |:---:|:---:|:---:|:---:|:---:|:---:|
> > > > | **Model Architecture** |  |  |  |  |  |
> > > > | Free-form generation | N | N | N | N | **Y** |
> > > > | Generalization | N | N | N | N | **Y** |
> > > > | Interpretability | N | N | N | N | **Y** |
> > > > | **Training data** |  |  |  |  |  |
> > > > | Diverse data | N | N | Y | N | **Y** |
> > > > | Realistic data | Y | N | Y | Y | **Y** |
> > > > | Faithful data | Y | Y | N | Y | **Y** |
> > > >
> > > > **Model Architecture**
> > > > * **Free-form generation**. [a,b,c,d] focus on improving compositional reasoning ability for CLIP or CLIP-like two-tower model. Their methods rely on a contrastive learning framework and cannot directly apply to free-form auto-regressive generative models. This limits their downstream tasks to retrieval-based tasks and is unable to apply to a broader form of tasks such as image captioning, dialogue, and visual question answering. Our method can be naturally applied to any auto-regressive generative model, thus able to do both free-form generation and retrieval-based tasks.
> > > > * **Generalization**. [a,b,c,d] focus on evaluating a single form of task, i.e., retrieving the correct caption from two candidate captions for a given image. This limits their generalization to more diverse tasks. For our method, thanks to the natural integration with auto-regressive LLM, we can generalize well to various tasks such as human-object interaction detection, referring expression comprehension, image caption, and visual question answering. The broadly applicable tasks make our method more general and bring more possibility to future work on diverse topics.
> > > > * **Interpretability**. Compared to [a,b,c,d], thanks to the grounding ability of our method, our method can visualize the compositional reasoning process. For example, if you prompt our model "the man is holding", our model will first generate the \<previsual\> token to locate the ROI for the potential object that the man is holding. It will probably choose the region containing the man's hand as the ROI since the object that the man is holding is most likely to be presented on the man's hand. By doing so, we can **visualize** relation by turning the abstract action into a visible region. The visual entity in the image can also be visualized using \<visual\>. The visualizability and interpretability of the compositionality in our method are unique and helpful in situations that need the model to be more interpretable.

---

> ### Author Response · Authors · 2023-11-21
> **Followup (part2)**
>
> **Training data**
> * **Diverse data**. Some previous works such as [a,b,d] rely on rule-based templates to create specific training data. MosaiCLIP [a] uses a scene graph generator to generate the scene graph of the image, then construct captions based on this scene graph information using simple templates. SyViC [b] utilized synthetic data generated by a simulation platform. They then compiled a list of sentences, including a caption prefix, object enumeration, pairwise positional relations between objects, scene description, and action and clothing descriptions for each human. The sentences were concatenated to form a complete dense caption for the image. TSVLC [d] first parses the text into components such as nouns, verbs, etc., and replaces one word in that text using either pre-defined word lists or LLM prediction to form new captions. Our method relies on raw caption text and can be applied to any large-scale image-text pair datasets using an automatic labeling pipeline, so we have the maximum diversity in data.
> * **Realistic data**. SyViC [b] relies on synthetic data, which may induce a domain gap and lack of variety in content compared to using real-world image-text data. Our method can use data from any image-text data source.
> * **Faithful data**. Faithful data means that the text should be related to the image and there are no hallucinations in data or introduced during the data preprocessing. DAC [c] relies on LLM to generate more captions for a given image and its ground truth caption by asking LLM "What should I expect to see in an image of \{caption\}?". Some facts that generated by LLM are likely to be hallucinated and unrelated to the image since the LLM cannot see the image directly. Therefore, unfaithful data is generated for training which will harm the model. Unlike DAC, we keep the raw caption, and the grounding process will only ground the objects that are in the image to the textual description in the text, doing our best to reduce hallucinations.
>
> Since only TSVLC releases their pre-trained weight, we choose to compare the compositional reasoning ability of CLIP, TSVLC, and our model using the ARO Top-1/5 metric to further illustrate the importance of diverse, realistic, and faithful data.
>
> | Method | ARO Top-1 | ARO Top-5 |
> |---|---|---|
> | CLIP | 6.93 | 21.12 |
> | TSVLC | 8.19 | 22.78 |
> | **Ours** | **32.46** | **55.70** |
>
> The results show that although TSVLC can improve the compositional reasoning ability for CLIP, our method can perform much better, thanks to the diverse, realistic, and faithful training data we have.
>
> We thank for the Reviewer pointing out the overclaim issue, and we have already toned down this statement in our revision. Nevertheless, our method has significant differences from previous works as analyzed above. We test our method on various datasets and tasks including ARO, Cola, HICO-DET, RefCOCO, and VQAv2, and get significant performance improvement, showing the effectiveness of our method. The simple but effective nature of our method is also helpful when scaling up the model or pre-training data, leading to even better performance and wider application for much larger LLMs.
>
> > **Reference:**
>
> [a] Coarse-to-Fine Contrastive Learning in Image-Text-Graph Space for Improved Vision-Language Compositionality. ICCV 2023.
>
> [b] Going Beyond Nouns With Vision & Language Models Using Synthetic Data. ICCV 2023.
>
> [c] Dense and Aligned Captions (DAC) Promote Compositional Reasoning in VL Models. NeurIPS 2023.
>
> [d] Teaching Structured Vision & Language Concepts to Vision & Language Models. CVPR 2023.

---

> > ### Comment · Reviewer_fA94 · 2023-11-22
> > **Thanks for the response**
> >
> > I value the authors' efforts in introducing new related work and comparisons. However, I maintain that the current solution, while an improvement, is somewhat simplistic and lacks a fundamental understanding of the problem.
> >
> > In recognition of their efforts, I've increased my score to 8, but this does not imply the paper is flawless.

---

### Official Review · Reviewer_Aw71 · 2023-10-30

**Soundness:** 3 good
**Presentation:** 4 excellent
**Contribution:** 3 good
**Rating:** 6
**Confidence:** 4

**Summary:**

The paper presents Compositional VLM, a vision-language architecture allowing the language model (LLM) to communicate with the vision encoder(s) to generate its output iteratively. Several new tokens, such as <box>, <prebox>, <visual>, and <previsual>, are introduced to facilitate this communication between the two components working on different modalities. The goal is to improve the ability of VLMs to capture the compositional structure found in vision-language tasks. The model is pre-trained on a large corpus of grounded image-text pairs from sources such as COCO, CC3M, CC12M, Visual Genome (Krishna et al., 2017), SBU, and LAION400M. The model is thoroughly evaluated on many tasks, such as compositional visual understanding, referring expression comprehension, VQA, and human-object interaction. The quantitative performance shows the benefit of improved communication between vision-language modules and outperforms considered VLM baselines on public benchmarks.

**Strengths:**

+ The paper addresses an important topic for VLMs. Imbuing VLMs with compositional visual understanding is an important direction of research, and the proposed approach is an interesting mechanism to achieve it.
+ The bidirectional communication between the vision and language modules is interesting and is an important component that needs to be introduced and explored in VLM research. Existing works do not have iterative communication, and relying only on global features for alignment/pre-training is insufficient to exhibit compositional understanding.
+ The quantitative performance is strong and shows consistent gains over the considered baseline VLMs on several tasks and benchmarks.
+ The paper is well-written and the idea is simple and presented intuitively, making it easy to follow and understand.

**Weaknesses:**

- My primary concern is about the pre-training data and how to interpret the results. The pre-training dataset consists of text-image pairs from COCO, CC3M, CC12M, Visual Genome (Krishna et al., 2017), SBU, and LAION400M. The evaluation datasets share considerable overlap with the pre-training data. For example, COLA is composed of data from GQA (derived from Visual Genome), CLEVR, and LVIS (also includes COCO). Similarly, ARO is based on Visual Genome, and HICO-DET is based on Flickr (with objects limited to those from COCO). I understand that the training data is essentially the same as BLIP-2 and that it is common for VLMs to be trained on these datasets. The question does remain: given this overlap, how well does the model generalize? The generalization can be quantified based on two factors - tasks and domains. From the task perspective, the generalization of the proposed approach seems to be limited, going by the performance of VQA. There are no convincing arguments for the generalization beyond training data. How about the performance on PACO[1]? It does not have a **significant** overlap with the pre-training datasets and should provide some measure of generalization.
- There are very few qualitative results presented. There are 4 examples in the supplementary, but they are very limited beyond that. By reading the paper, it is hard to understand when/where/why the model fails. For example, is there a reason why the model has a higher performance on the "rare" class from HICO-DET? It would be good to understand the success and failure modes and have a discussion on the approach's qualitative performance as opposed to the purely quantitative take.
- There are no ablations presented. What does the choice of encoders/pre-training data/tokens have on the model?
- On that note, how sensitive is the model to the template used for prompting?

References:
[1] Ramanathan, Vignesh, et al. "Paco: Parts and attributes of common objects." Proceedings of the IEEE/CVF Conference on Computer Vision and Pattern Recognition. 2023.

**Questions:**

My major concerns are the generalization capabilities and the lack of qualitative results and ablation studies, detailed in the weaknesses section.

--- Post-rebuttal update ---
I am raising my score after the authors' response.

---

> ### Author Response · Authors · 2023-11-16
> **Response to Reviewer Aw71**
>
> We thank you for your time and valuable comments. Below we answer the main concerns raised in the review and would be happy to provide further clarification if suitable.
>
> > **Q1. Generalization capabilities.**
>
> We thank the reviewer for recommending a related benchmark PACO [a] that is more suitable to provide some measure of generalization. We evaluate the AR@1 metric on all L1 queries under the zero-shot instance detection setting.
>
> | Model | Type | PACO L1 AR@1 |
> |---|---|---|
> | MDETR R101 [b] | Open-vocabulary detector | 4.9 |
> | Detic Swin-B [c] | Open-vocabulary detector | 5.9 |
> | KOSMOS-2 | VLM | 8.2 |
> | Compositional VLM 1.4B | VLM | 9.4 |
>
> The results show that our method can generalize well to other image domains and harder tasks. Please see G3 for a detailed explanation of the results.
>
>
> > **Q2. There are very few qualitative results presented..**
>
> Thanks. We have added more qualitative results and discussion in Appendix A.5 in our revision.
>
> > **Q3. There are no ablations presented. What does the choice of encoders/pre-training data/tokens have on the model?**
>
> Thanks for your advice. We have added an ablation study on communication tokens in G1.
>
>
> | Setting | <previsual> | <prebox> | <visual> | <box> | ARO Top-1 | Cola |
> |:---:|:---:|:---:|:---:|:---:|:---:|:---:|
> | 1 | x | x | x | x | 29.60 | 38.10 |
> | 2 | x | x | ✓ | ✓ | 30.08 | 34.76 |
> | 3 | ✓ | x | ✓ | x | 32.26 | 36.19 |
> | 4 | ✓ | ✓ | ✓ | ✓ | **32.46** | **44.29** |
>
> The ablation study results demonstrate the necessity and effectiveness of the proposed communication tokens. Please refer to G1 for detailed explanation.
>
> For the choice of encoder, we follow previous works such as BLIP-2 [d] and LLaVA [e] to use pre-trained CLIP ViT-L [f] as our vision encoder, as the pre-trained CLIP ViT is proved to be a powerful model in various downstream tasks. For the choice of pre-training data, the only concurrent public large-scale grounded image-text dataset is from KOSMOS-2, but KOSMOS-2 only releases a 20 million subset of its full 90 million data. As the data quantity is important for multimodal pre-training, we choose to create our own large-scale grounded image-text dataset based on the widely used BLIP-2 pre-training data. Our pre-training dataset consists of over 97 million image-text pairs, comparable to KOSMOS-2's full pre-training data.
>
> Due to the short period for rebuttal and the time-consuming nature of multimodal pre-training, it is hard for us to conduct an ablation study on encoders and pre-training data during the rebuttal period. We leave the additional ablation study on different encoders and different sets of pre-training data in our future work and we are happy to have more discussion with the Reviewer if needed.
>
> > **Q4. How sensitive is the model to the template used for prompting?**
>
> Our model is robust to the template used for prompting. We evaluate the performance of using different prompts in RefCOCOg/RefCOCO+/RefCOCO benchmarks.
>
> | Prompt | RefCOCOg val | RefCOCO+ val | RefCOCO val |
> |---|:---:|:---:|:---:|
> | Locate the object: <obj>{text}</obj><visual> | 60.64 | 46.83 | 47.50 |
> | Locate: <obj>{text}</obj><visual> | 60.89 | 46.97 | 47.63 |
> | Find: <obj>{text}</obj><visual> | 60.89 | 47.20 | 47.60 |
> | <obj>{text}</obj><visual> | 60.87 | 47.62 | 48.19 |
>
> The evaluation results do not differ much when using different prompts, indicating that our model is robust to the prompt template.
>
> > **Reference:**
>
> [a] PACO: Parts and Attributes of Common Objects. CVPR 2023.
>
> [b] MDETR - Modulated Detection for End-to-End Multi-Modal Understanding. ICCV 2021.
>
> [c] Detecting Twenty-thousand Classes using Image-level Supervision. ECCV 2022.
>
> [d] BLIP-2: Bootstrapping Language-Image Pre-training with Frozen Image Encoders and Large Language Models. ICML 2023.
>
> [e] Visual Instruction Tuning. NeurIPS 2023.
>
> [f] Learning Transferable Visual Models From Natural Language Supervision. PMLR 2021.

---

> > ### Author Response · Authors · 2023-11-21
> >
> > Dear reviewer Aw71,
> >
> > We really appreciate your comments to strengthen this work. As the rebuttal period is ending tomorrow, we wonder if our response convince you. If yes, would you kindly consider raising the score? Thanks again for your very constructive and insightful feedback!
> >
> > Best,
> >
> > Authors

---

> > ### Comment · Reviewer_Aw71 · 2023-11-21
> > **Thanks for the response**
> >
> > Thanks for your response with the additional comparison. It does mostly address my concerns. I am raising my score to 6.

---

### Official Review · Reviewer_onDF · 2023-11-01

**Soundness:** 3 good
**Presentation:** 2 fair
**Contribution:** 3 good
**Rating:** 5
**Confidence:** 4

**Summary:**

This paper introduces Compositional VLM, a new Large Vision-language Model that can compose visually grounded concepts and relationships within a given text input. It achieves this by employing a set of special tokens to manipulate the interactions between the LLM and a visual object detection network. This bi-directional communication of vision-to-language and language-to-vison occurs through multiple iterations as an output sentence is generated. The proposed model outperformed prior works across various compositional reasoning tasks and vision-language tasks.

**Strengths:**

- The object-centric approach presented in this paper is interesting as I believe visual objects and words are of the same level of abstraction of representation. I also appreciate the idea of multi-step interactions between an LLM and a visual object detection network, much like older multi-step reasoning techniques in the joint vision-language understanding literature.

- The experiments demonstrate significant improvements over the existing works across a number of downstream tasks and benchmarks.

**Weaknesses:**

- The presentation of the method is brief, making it hard to understand. It is not clear how the communication tokens are generated. I recommend providing a pseudo algorithm to describe the method and making the input/output for each step more readable.

- The proposed pre-training procedure seems to incorporate a lot of engineering tricks and data processing techniques from KOSMOS-2, without adequate attribution to previous works. The paper also lacks justifications for the selection of sub-components in the proposed method, making it hard to interpret the results.

- Pretrained data and downstream tasks (i.e., RefCOCO/RefCOCO+, VQA v2) share the same pool of visual data taken from COCO and Visual Genome. This raises concerns about the validity of the reported performance on these downstream tasks.

**Questions:**

-	Why did you opt to create your own dataset of similar scale for pretraining, rather than utilizing the data from KOSMOS-2?
-	Could you provide further analyses of the contributions of these communication tokens?
-	On the task REFERRING EXPRESSION COMPREHENSION: the proposed model outperforms the KOSMOS-2 model, but with a small gap. While the grounding of vision-language in KOSMOS-2 was reasonably good compared to the proposed model, it exhibited significantly poorer performance in the previous three tasks. Can you provide insights into these differences?

-	Can you clarify the large margins between KOSMOS-2 and the proposed method in Table 1? What would be the main contributor to this significant performance leap?

---

> ### Author Response · Authors · 2023-11-16
> **Response to Reviewer onDF (part1)**
>
> We thank you for your time and valuable comments. Below we answer the main concerns raised in the review and would be happy to provide further clarification if suitable.
>
> > **Q1. Why do you create your own dataset rather than using data from KOSMOS-2.**
>
> KOSMOS-2 has made their data public, but they have only released a **subset** of it, about 20 million image-text pairs instead of the full dataset with 90 million image-text pairs. A large-scale and high-quality dataset is essential for pre-training, therefore, we turned to create our own dataset which has a similar scale compared to KOSMOS-2. The dataset we create consists of over 97 million image-text pairs. We will make our full dataset public in the future.
>
> > **Q2. Could you provide further analyses of the contributions of the communication tokens?.**
>
> Communication tokens are designed to enable bidirectional communication between the vision module and the language module. They can summarize the compositionality information within text to guide the vision module to detect the desired objects, and can also bring the visual feature of the object back to the language module to guide the LLM to notice the localized visual feature, thus helping it to better generate more faithful and related tokens thereafter. We have added an ablation study on communication tokens in G1.
>
> | Setting | <previsual> | <prebox> | <visual> | <box> | ARO Top-1 | Cola |
> |:---:|:---:|:---:|:---:|:---:|:---:|:---:|
> | 1 | x | x | x | x | 29.60 | 38.10 |
> | 2 | x | x | ✓ | ✓ | 30.08 | 34.76 |
> | 3 | ✓ | x | ✓ | x | 32.26 | 36.19 |
> | 4 | ✓ | ✓ | ✓ | ✓ | **32.46** | **44.29** |
>
> The ablation study results demonstrate the necessity and effectiveness of the proposed communication tokens. Please refer to G1 for detailed explanation.

---

> ### Author Response · Authors · 2023-11-16
> **Response to Reviewer onDF (part2)**
>
> > **Q3. The proposed model outperforms the KOSMOS-2 model in referring expression comprehension tasks but with a small gap. While the grounding of vision-language in KOSMOS-2 was reasonably good compared to the proposed model, it exhibited significantly poorer performance in the previous three tasks. Can you provide insights into these differences?**
>
> The performance gap is small in referring expression comprehension tasks because KOSMOS-2 and our model both adopt a similar manner for testing. This gap is large in the previous three tasks because the communication tokens in our method can facilitate our model to understand object relations and focus on a localized ROI to generate more faithful object predictions. The deeper analysis is as below:
>
> 1. For **referring expression comprehension** tasks, both of us use a format of *\<obj\>*{text}*\</obj\>* as the prompt to the model, and then enable the model to output location information. KOSMOS-2 outputs some location tokens to represent the location, while our model outputs the bounding box through the object detector. The advantages of communicative decoding and the bidirectional communication between the vision module and the language module are not fully exploited in this task, thus the performance gap between KOSMOS-2 and ours is reasonably small.
>
> 2. The **HICO-DET** benchmark asks the model to detect all human-object interactions in the image. Many images contain more than one human-object interaction, that is, there are multiple objects corresponding to the same textual description that need to be identified correctly. We empirically find that compared to KOSMOS-2, our model will make it easier to detect more correct objects corresponding to the same correct textual description in the image. Therefore, our model can find out the more complete human-object interactions within an image. This is thanks to our YOLO-like object detector that can naturally identify all correct objects in the image. The input image will be encoded by the vision encoder to become a map of 16x16 patches. When performing the object detection, each patch will be concatenated with the hidden state of *\<previsual\>*/*\<visual\>* from LLM, and then fed to the object detector to get a score indicating the probability that this patch contains the center of the object, and also get a regressed bounding box coordinates indicating the left-top and right-bottom coordinates of the object if exists. Each patch can produce a probability and object bounding box individually, thus it is natural and convenient for our model to identify multiple objects.
>
> 3. **ARO** requires that given entity A and relation, the model should predict entity B to which the relation is applied. It requires the model to understand where is entity A and what's the relation and use this information to infer the entity that A is applying the relation to. The design of *\<previsual\>*/*\<prebox\>* communication tokens can handle this relation information well. By using the information of entity A and the relation, *\<previsual\>*/*\<prebox\>* communication tokens can guide the LLM to focus on the potential regions for entity B, so it would be much easier to correctly predict entity B. **Cola** benchmark also require the model to understand both the objects and their relation, thus *\<previsual\>*/*\<prebox\>* communication tokens can help in this task. Since KOSMOS-2 lacks communication tokens and explicit guidance for LLM to focus on a specific region, it does not perform well in such tasks.
>
> > **Q4. Can you clarify the large margins between KOSMOS-2 and the proposed method in Table 1? What would be the main contributor to this significant performance leap?**
>
> The large margins between KOSMOS-2 and our proposed method in Table 1 are mainly because our model has adopted a novel bidirectional communication between the vision module and language model via communication tokens, which brings a large improvement in compositional reasoning ability. Our ablation study especially shows the importance of the proposed communication tokens for compositional tasks (please refer to G2). KOSMOS-2 does not have this kind of design and thus performs much poorer in these tasks.
>
> > **Q5. The presentation of the method is brief, making it hard to understand. It is not clear how the communication tokens are generated. I recommend providing a pseudo algorithm to describe the method and making the input/output for each step more readable.**
>
> Communicate tokens are automatically generated during the auto-regressive generation process. We have added an explanation of the communication token generation in G1 and a pseudo algorithm in Appendix A.1 in our revision.

---

> ### Author Response · Authors · 2023-11-16
> **Response to Reviewer onDF (part3)**
>
> > **Q6. No adequate attribution to previous works.**
>
> Thanks for your reminder. During our data-creating process, we have borrowed some techniques from KOSMOS-2 including using spaCy to parse the text and use *\<obj\>\</obj\>* to enclose an object. We also have some differences from KOSMOS-2. For example, we use GroundingDINO to directly ground the textual description in the text to the object in the image. In contrast, KOSMOS-2 first parses the sentence to extract all noun chunks. Then, they eliminate certain abstract noun chunks that are hard to recognize such as "time" and "love". Finally, they use GLIP to obtain the bounding boxes for these noun chunks. In comparison, our data-creating method is more simple but still effective. We have revised to give more adequate attribution to previous works in our revision.
>
> > **Q7. Lack of justifications for the selection of sub-components in the proposed method.**
>
> Thanks for your reminder. We have added the ablation study on communication tokens. Detailed results and analysis are shown in G2, which indicates the effectiveness and necessity of these tokens.
>
> > **Q8. Pretrained data and downstream tasks (i.e., RefCOCO/RefCOCO+, VQA v2) share the same pool of visual data taken from COCO and Visual Genome.**
>
> It is worth noting that in KOSMOS-2's instruction tuning stage, LLaVA-Instruct [b] data is used to fine-tune the model, and the images in LLaVA-Instruct data are from COCO. Therefore, KOSMOS-2's training data and downstream tasks also share the same pool of visual data from COCO. The visual data overlap concern is not unique to our model.
>
> To prove the generalization of our model, we further evaluate our model on PACO [a] as recommended by reviewer Aw71.
>
> | Model | Type | PACO L1 AR@1 |
> |---|---|---|
> | MDETR R101 [b] | Open-vocabulary detector | 4.9 |
> | Detic Swin-B [c] | Open-vocabulary detector | 5.9 |
> | KOSMOS-2 | VLM | 8.2 |
> | Compositional VLM 1.4B | VLM | 9.4 |
>
> Our model continues to perform well in this dataset, indicating our model's good generalization to other image domains. The dataset description and result analysis for the PACO dataset can be found in G3.
>
> > **Reference:**
>
> [a] PACO: Parts and Attributes of Common Objects. CVPR 2023.
>
> [b] MDETR - Modulated Detection for End-to-End Multi-Modal Understanding. ICCV 2021.
>
> [c] Detecting Twenty-thousand Classes using Image-level Supervision. ECCV 2022.

---

> > ### Author Response · Authors · 2023-11-21
> >
> > Dear reviewer onDF,
> >
> > Thanks again for your suggestion to strengthen this work. As the rebuttal period is ending soon, we wonder if our response answers your questions and addresses your concerns. If yes, would you kindly consider raising the score? Thanks again for your very constructive and insightful feedback!
> >
> > Best,
> >
> > Authors

---

> > > ### Author Response · Authors · 2023-11-22
> > > **Reminder for Reviewer onDF**
> > >
> > > Dear reviewer onDF:
> > >
> > >
> > > Thanks again for your suggestion to strengthen this work. As the rebuttal period is ending tomorrow, we wonder if our response answers your questions and addresses your concerns. If yes, would you kindly consider raising the score? Thanks again for your very constructive and insightful feedback!
> > >
> > > Best,
> > >
> > > Authors

---

### Author Response · Authors · 2023-11-16
**General Response to All Reviewers and ACs (part1)**

We would like to thank all reviewers and ACs for their time and efforts in reviewing our paper and giving insightful comments. We are glad that the reviewers have recognized our contributions listed below:

* **Method**. Communicative decoding is a novel multi-step and bidirectional communication mechanism between the vision and language modules during text generation (onDF and Aw71). With well-designed communication tokens, it can imbue VLMs with compositional visual understanding (Aw71) and bridge the gap between visual entities and their textual descriptions (fA94).
* **Experiments**. The improvement on various downstream tasks is significant (onDF) and shows consistent gains (Aw71).


We also appreciate reviewers' constructive suggestions and concerns. Besides the response to individual reviewers, we explain some general concerns and provide a brief description of the revisions we have made.

> **G1) How the communication tokens are generated?**
>
The communication tokens are **automatically generated by LLM** or **automatically inserted following each other**. Specifically, for *\<previsual\>* and *\<visual\>* tokens, they are automatically generated by LLM through the next token prediction. For *\<prebox\>* and *\<box\>* tokens, they are automatically inserted following *\<previsual\>* or *\<visual\>* tokens, respectively.

It is worth noting that the communication token prediction process is compatible with auto-regressive generation and thus can be seamlessly applied to concurrent powerful LLMs, enhancing their compositional reasoning ability. We have added a pseudo algorithm in Appendix A.1 in our revision to help illustrate the generation process.

---

> ### Author Response · Authors · 2023-11-16
> **General Response to All Reviewers and ACs (part2)**
>
> > **G2) Ablation study on the communication tokens.**
> >
> In our paper, we have designed four special tokens for communication between language and images, namely *\<previsual\>*, *\<visual\>*, *\<prebox\>*, and *\<box\>*. According to their function, we divide them into two categories:
> * *\<previsual\>* and *\<visual\>* summarize the text information so far, and send the summarized information to the object detector to detect ROIs;
> * *\<prebox\>* and *\<box\>* contain the visual feature for the detected ROI and are put back into the language sequence to make LLM notice these localized visual features for better generating future tokens.
>
> To evaluate the effectiveness of these communication tokens, we conduct an ablation study on Compositional VLM 1.4B model. We design four settings:
>
> * To evaluate the overall effectiveness of communication tokens, we designed setting 1, which has no communication token.
> * To evaluate the effectiveness of putting grounding process right after the object (this is similar to what KOSMOS-2 does), we design setting 2, which adds *\<visual\>* and *\<box\>* to enable the model to predict the object's location after the object is generated.
> * Based on setting 2, to evaluate the effectiveness of *\<previsual\>* and *\<prebox\>*, we further add *\<previsual\>* and *\<prebox\>*. With *\<previsual\>*, the model can predict the ROI of the object based on the former object and the relation information. With *\<prebox\>*, the model can put the localized visual feature of the ROI back into the sequence before generating the object. This forms setting 4, which also represent the full set of communication tokens.
> * To evaluate what's the impact of sending the localized visual feature back to LLM, we design setting 3, which eliminates *\<prebox\>*/*\<box\>* so that the model will only detect the ROI but not put the visual feature of the ROI back into LLM.
>
> | Setting | <previsual> | <prebox> | <visual> | <box> | ARO Top-1 | Cola |
> |:---:|:---:|:---:|:---:|:---:|:---:|:---:|
> | 1 | x | x | x | x | 29.60 | 38.10 |
> | 2 | x | x | ✓ | ✓ | 30.08 | 34.76 |
> | 3 | ✓ | x | ✓ | x | 32.26 | 36.19 |
> | 4 | ✓ | ✓ | ✓ | ✓ | **32.46** | **44.29** |
>
> The results in the table above reveal that our proposed communication tokens play a critical role in compositional reasoning tasks, improving ARO from 29.60 to 32.46, and Cola from 34.76 to 44.29. Going deeper into these results, we have the following insights:
>
> 1. By comparing setting 1 and setting 4, we can see that with no communication token, our model's performance on compositional reasoning tasks is similar to BLIP-2. This is reasonable because we share the same pre-training data with BLIP-2 and the additional grounding information is not used during training.
> 2. By comparing setting 1 and setting 2, we can find that generate *\<visual\>*/*\<box\>* **after** the predicted object does not help compositional reasoning. We hypothesize that it is because adding extra information **after** the predicted object description won't help the generation of that object description itself as this is an auto-regressive generation process: the context after one certain token won't have an impact on the generation of that certain token.
> 3. By comparing setting 2 and setting 4, we can notice that adding *<previsual>* and *<prebox>* **before** the object that will be predicted can boost the performance on both ARO and Cola. We hypothesize the reason is that the ROI prediction and the inserted localized visual feature **before** the object that are about to be predicted will help LLM to better generate that object description. This hypothesis is based on a simple but important rule: the context before one certain token will have an impact on the generation of that certain token in auto-regressive generation.
> 4. By comparing setting 3 and setting 4, we can conclude that not putting visual features of ROIs, represented by *\<prebox\>*/*\<box\>*, back into the generated sequence will hurt the compositional reasoning ability for complex object description. ARO Top-1 accuracy does not hurt much if we do not put *\<prebox\>*/*\<box\>* back, while the performance for Cola will drop significantly. We speculate this is because the predicted object in ARO is usually a simple word, while the object in Cola is a complex phrase. In this case, *\<prebox\>*/*\<box\>* tokens which contain more fine-grained information about the visual entity can play an important role in assisting LLM to focus on these complex phrases and generate more faithful and related tokens thereafter.

---

> ### Author Response · Authors · 2023-11-16
> **General Response to All Reviewers and ACs (part3)**
>
> > **G3) How well does the model generalize to the dataset that does not overlap with the training data?**
>
> To further evaluate the generalization ability of our model, we conducted further experiments on PACO [a] recommended by reviewer Aw71. The images for this dataset come from LVIS [b] and Ego4D [c], so it does not have a significant overlap with our training data. As a result, it can provide some measure of our model's generalization.
>
> PACO is a difficult task for existing methods because it requires the model to identify whether the object in the image is **exactly** the same as the query. For example, if the query is a long wooden chair, then the long plastic chair in a given image should not be grounded. There is only one positive image that contains the query object, while 100 negative images, containing an object very similar to the query object, are presented as distractors.
>
> We evaluate the AR@1 metric on all L1 queries under the zero-shot instance detection setting. MDETR R101 [d] and Detic Swin-B [e] are two baselines presented in PACO benchmark paper.
>
> | Model | Type | PACO L1 AR@1 |
> |---|---|---|
> | MDETR R101 [d] | Open-vocabulary detector | 4.9 |
> | Detic Swin-B [e] | Open-vocabulary detector | 5.9 |
> | KOSMOS-2 | VLM | 8.2 |
> | Compositional VLM 1.4B | VLM | 9.4 |
>
>
> The evaluation results show that even though this task is hard, our model performs better than other zero-shot baselines. The result strongly supports the generalization of our model to datasets in other domains.
>
> **Revision Summary**
>
> We made the following modifications in our revision to address reviewers' questions (highlighted in blue in the revision):
> 1. We provide a pseudo algorithm in Section A.1 to illustrate how the communication tokens are generated (onDF and fA94).
> 2. We conduct ablation studies on communication tokens in Section 4.3 (onDF and Aw71).
> 3. We present more qualitative results and analysis in Section A.5 (Aw71).
> 4. We add more clear attribution to previous works for our data-creating pipeline (onDF).
>
> > **Reference:**
>
> [a] PACO: Parts and Attributes of Common Objects. CVPR 2023.
>
> [b] LVIS: A Dataset for Large Vocabulary Instance Segmentation. CVPR 2019.
>
> [c] Ego4D: Around the World in 3,000 Hours of Egocentric Video. CVPR 2022.
>
> [d] MDETR - Modulated Detection for End-to-End Multi-Modal Understanding. ICCV 2021.
>
> [e] Detecting Twenty-thousand Classes using Image-level Supervision. ECCV 2022.

---

### Author Response · Authors · 2023-11-21
**Thank you and we are looking forward to your post-rebuttal feedback!**

Dear AC and all reviewers:

Thanks again for all the insightful comments and advice, which helped us improve the paper's quality and clarity.

The discussion phase has been on for several days and we are still waiting for the post-rebuttal responses.

We would love to convince you of the merits of the paper. Please do not hesitate to let us know if there are any additional experiments or clarification that we can offer to make the paper better. We appreciate your comments and advice.

Best,

Author

---

### Meta-Review · Area_Chair_PtEK · 2023-12-19

**Metareview:**

Building more compositional reasoning into vision-language models is of wide interest to the community. The method presented, as the reviewers point out, is not what one might traditionally think as compositional. It enables the VLM to communicate with the vision encoder and detection network more effectively. In a sense this is compositional, and it certainly improves performance on compositional benchmarks (although it may also just have higher performance in general). The technique used could be applied to other domains, like robotics, as it essentially involves giving the model as input a kind of markup language to help it reason about its available resources.

Reviewers pushed back against the datasets used and wanted experiments further afield as well as ablations. Authors provided both and showed convincingly that the method works well even on datasets and tasks they had not considered earlier; a big plus.

This idea is likely to be of interest to many in the ICLR community. Those interested in neurosymbolic reasoning are likely also going to be interested as well. Perhaps that would have been an alternative pitch for the method.

**Justification For Why Not Higher Score:**

The limited notion of compositionality, but certainly markup languages to help models reason are interesting and a useful contribution.

**Justification For Why Not Lower Score:**

A useful approach to an important problem.

---

### Decision · Program_Chairs · 2024-01-16

Accept (poster)